# Toward Net-Zero: The Barrier Analysis of Electric Vehicle Adoption and Transition Using ANP and DEMATEL

**Tsai-Chi Kuo** [1,2] **, Yung-Shuen Shen** [3,*]**, Napasorn Sriwattana** [1] **and Ruey-Huei Yeh** [1]

1   Department of Industrial Management, National Taiwan University of Science and Technology, Taipei 10607, Taiwan
2   Artificial Intelligence for Operations Management Research Center, National Taiwan University of Science and Technology, Taipei 10607, Taiwan
3   Institute of Geriatric Welfare Technology & Science, Mackay Medical College, New Taipei City 25245, Taiwan
*   Correspondence: ysshen@mmc.edu.tw

**Abstract:** Global greenhouse gas emissions must be reduced to achieve net-zero carbon emissions. One of the solutions for the reduction of greenhouse gas emissions is the adoption and transition from conventional vehicles to electrical vehicles (EVs). Previously, most research on EVs have been from a consumer adoption perspective, few of them are from industry transition and consumer adoption perspectives simultaneously. This also highlights the importance of SDG 12 (responsible for consumption and production). Additionally, the analyses were mostly obtained using one methodology and demonstrated only by weighting without relationships among factors. To consider the problem of adoption and transition, a systematic method should be developed. Therefore, this study intends to identify, prioritize, and display the relationship between EV adoption barriers from an automotive industry perspective using an analytic network process (ANP) and the decision-making trial and evaluation laboratory (DEMATEL) method. The research results show two contributions: First, the identified and prioritized barriers that automakers encounter in EV transition also explored the interrelationships among these barriers. Second, a model comparison of two multicriteria decision-making approaches was conducted to prioritize and identify the interlinkages among EV uptake barriers.

**Keywords:** electrical vehicle transition; multi-criteria decision making; ANP; DEMATEL; SDG 12





## 1. Introduction

Climate change caused by migration, which is widely acknowledged by scientists, is driven by an increased level of greenhouse gases (GHGs). According to the Paris Agreement [1], many governments from different countries have set a target to achieve net-zero by 2050 [2]. Net-zero considers total emissions, permitting the elimination of any inevitable emissions, such as those from transportation or industry.

To achieve net-zero emissions [3], GHG emissions must be reduced rapidly and significantly, and removal must be scaled up [4]. The transportation sector contributes over 16.2% to worldwide GHG emissions, and this percentage is likely to rise in the future [5]. Despite breakthroughs in biofuels and electricity, transportation has generally been based on oil fuels, which accounted for more than 90% of the transport sector's power needs in 2020 [6].

Williams et al. [7] suggested the electrification of transportation required to meet an 80% reduction target. Additionally, an adaptation from internal combustion engine (ICE) vehicles to higher levels of sustainable transportation could be a major contributor towards decreasing global GHG emissions [8]. Electric vehicles (EVs) are emerging as an important substitute solution with conventional vehicles for transportability, contributing to the largest share of transportation emissions [8]. If efficiently and consistently implemented, electric vehicle usage will lead to significant decarbonization, and lead to improved environmental quality as a result, particularly if EVs are combined with a low-carbon electricity

generation system [9]. Furthermore, EVs also provide several advantages besides decarbonization, such as no tailpipe emissions, thereby preventing air pollution and exposure to volatile organic compounds, nitrogen oxides, and carbon monoxide in residential areas [10].

Many countries are currently working to lower emissions from the transportation sector. In 2019, governments worldwide expended approximately USD 14 billion to promote the sales of electric vehicles, up 25% from the previous year, potentially resulting in greater incentives in Europe. However, over the last five years, the share of government subsidies in total spending on electric cars has declined, demonstrating that EVs are becoming more appealing to customers [3].

The desire for environmentally friendly vehicles to run on alternate fuels is increasing significantly. As a result, the automobile industry is rapidly transitioning to something that is more environmentally sustainable, and the world's main automobile makers, as well as many countries, are attempting to gain a competitive advantage in electric vehicles and innovative technology [11]. The increased sales trends for EVs in the foreseeable future are possible. In the first quarter of 2021, worldwide EV sales rose by approximately 140 percent compared to the same period in 2020, owing to the sales of around 500,000 cars in China and around 450,000 in Europe. From a lower foundation, sales in the United States more than doubled in the first quarter of 2020 [3].

Despite their potential usefulness, considerable obstacles or barriers to widespread EV diffusion technology remain, and EVs now represent only a small percentage of all vehicles. Various barriers, causes, and difficulties related to diffusion of electric vehicles have been identified and reported in previous studies [12–14]. She et al. [15] classified the barriers into three categories: economic, efficiency, and facilities scoping in China. A recent study [16,17] divided these barriers into five categories using the analytic hierarchy process (AHP).

EV transition is a significant global issue. Several industrial sectors should be carefully studied and supported to achieve EV transition. However, in earlier EV transition-related studies, only the consumer perspective was widely studied. In the case of successful EV transition, apart from focusing on consumer desire, the automotive industry perspective is also pivotal to the study. In previous research that focused on barriers to EV transition, the indicated and prioritized results were computed using only one methodology. Moreover, these outputs only display barrier categories and the weight of each barrier, but do not display a relationship between the barriers. However, the linkage between barriers or their influence on each other is necessary in real-world implementation, which existing research rarely identifies. This highlights the research goal of systematically evaluating the barriers to EV transition, not only from the perspective of consumers, but also EV companies. This paper is organized as follows:

(1) Identification of critical barriers that have the ability to influence electric vehicle transition from an automotive industry perspective.
(2) Analysis and prioritization of the barriers due to their particular effect on EV transition using the analytic network process (ANP) and decision-making trial and evaluation laboratory (DEMATEL) methodology.
(3) Analysis of the linkage between each barrier of EV transition.

The remainder of this paper is organized as follows: A literature review is presented in Section 2, Section 3 describes multi-criteria decision making (MCDM) solutions for the barriers to EV transition, and Section 4 provides the results. Finally, the conclusions and opportunities for future studies are presented in Section 5.

## 2. Materials and Methods

Despite progress so far, EV transition still has a long way to go before it reaches a large scale, which is dependent on the transformation of energies, technologies, and customer behaviors. According to the literature review, these include technologies regarding electric vehicles, energy transformation of electricity, and electric vehicle barriers.

## 2.1. Technology regarding Electric Vehicles (EVs)

Internal combustion engines (ICEs) in conventional cars (CVs) burn fossil fuels, operate ineffectively, and produce a large quantity of greenhouse gases. Alternative fuel vehicles (AFVs), including electric cars, fuel cell vehicles, and biofuel vehicles, are designed to run on at least one alternative to petroleum and diesel. Currently, there are three types of electric vehicles: battery electric vehicles (BEVs), plug-in hybrid electric vehicles (PHEVs), and hybrid electric vehicles (HEVs). Table 1 lists the categories of EVs. The driving range, charging duration, refueling costs, and engineering and design of each vehicle type varies significantly. In all countries, BEV models have been presented in most vehicle categories, and PHEVs have leaned towards larger vehicle segments. In all markets, sports utility vehicle (SUV) types contribute to half of the EVs models available [3].

**Table 1.** Electric vehicle category.

| Vehicle Type | Engine Description | Advantages | Disadvantages |
|---|---|---|---|
| Conventional | Internal combustion engine. | Relatively high-power motor and fast acceleration, starts quickly. | Emissions are generally high compared to external combustion engine. |
| Hybrid electric vehicles (HEVs) | Separated electric motor with internal combustion engine. | Compared to similarly conventional vehicles, has better fuel saving, cheaper operating costs, and lower pollutants. | Higher purchasing cost and complex hybrid technology. |
| Plug-in hybrid electric vehicles (PHEVs) | Powerful rechargeable battery, smaller internal combustion engine, and bigger electric motor. | Compared to HEVs and conventional vehicles, has better fuel efficiency, cheaper operating costs, and reduced pollutants. Also provides fuel source adaptability. | Not easy to maintain. |
| Battery electric vehicles (BEVs) | Chargeable battery packs which can be plugged into an electrical socket. | No liquid resources and pollutants. Compared to HEVs and conventional vehicles, it is less expensive to operate. | Battery waste generates environmental problems. |

EVs rely on electricity for part- or all of their propulsion, and they come in a variety of forms [12]. On the other hand, in hybrid electric vehicles (HEVs), a battery and an electric engine are added to a vehicle with an internal combustion engine (ICEs) to achieve better fuel efficiency than comparable-sized vehicles. Hybrid electric vehicles (HEVs) such as the Toyota Prius, Honda Insight, and Honda Civic Hybrid, have been commercially successful [18]. Plug-in hybrid electric vehicles (PHEVs) have a more efficient and bigger battery capable of energizing the car for approximately 20 to 60 miles, compared to HEVs. PHEVs reduce the size of the internal combustion engine to be smaller than that of HEVs. Moreover, the batteries of PEVs are rechargeable, and they can be fully charged by plugging into an external power source. PHEVs have superior fuel economy over EVs, but they also have the option of using conventional fuels for prolonged journeys [12]. Another alternative for EVs is battery electric vehicles (BEVs), which have been actively promoted because of their ability to lower local $CO_2$ and other hazardous air pollutants, while also reducing automobile noise. BEVs can significantly reduce emissions during operation, despite having higher emissions during manufacturing than ICEVs. BEVs are completely powered by a rechargeable electric battery and typically have larger batteries than PHEVs. They can reach up to 100 miles on a single charge [19].

## 2.2. Energy Transformation for Electricity

Gasoline price is a noticeable factor in EV adoption. Van Bree et al. [20] discovered that rising gas prices influence customers. Gallagher et al. [21] revealed that customers generally decide to purchase EVs because of higher fuel prices and government subsidies. Together with pricing, non-economic considerations, particularly those related to the en-

vironment and energy, can affect customers' alternatives to EVs. Moreover, according to Kahn [22], environmentalists are more likely to buy EVs than those who are not interested in environmental protection. Additionally, the study also identified that social preferences for environmental protection and energy policies were significant considerations in EV adoption [21].

During 2007–2008, gasoline price increases were caused by crude oil price rises and refining deficiency. An increased EV market share would immediately reduce fuel consumption, lessen petroleum refining shortages, and likely lower prices. Furthermore, consumers perceive that fuel prices will continue to increase in the years ahead. According to a survey, as fuel prices increase, more people believe that EVs have become a good investment [23].

Issues regarding rising GHG emissions and oil security have driven policymakers worldwide to prioritize low-carbon development and innovation for zero-carbon transportation. The prediction for increasing GHG emissions in the next 20 years is up to 45%, and this concern has compelled many states in the United States to accept regulations requiring a net-zero standard [24]. According to the IEA [3], the automotive sector accounts for 16.2% of global energy consumption and 25% of $CO_2$ emissions from energy-related sectors. If present trends persist, the worldwide transportation demand for energy and $CO_2$ emissions from energy is forecasted to double by 2050.

There is an issue that the electricity used to power electric vehicle is not completely clean since it mainly made from fossil fuels like coal and gas. Egbue [12] found that even electricity for EVs is not entirely carbon-free, but compared to conventional vehicles such as diesel or gasoline vehicles, EVs emit less than 50% GHG. Hassan [25] also illustrated that EVs that are fully powered by coal-refined electricity produce less than 25% GHG per mile compared to conventional vehicles.

*2.3. EV Adoption Barriers*

An obvious barrier is the high purchase price of EVs when considering conventional ICE vehicle prices. Additionally, from a consumer perspective, the total cost of ownership is also included in making purchase decisions, although when considering the total life cycle, EVs are less expensive than conventional vehicles. In addition, technical barriers are also a main factor that indicate the expansion of the EV market in the future. For example, battery performance is one of the most important factors, since electric vehicles are mostly powered by batteries, and consumers are concerned about the exact driving distance of EVs [26]. Charging time is also mentioned as a barrier to EV transition because users desire the shortest charging duration for daily travel [23]. Another major type of barrier are social barriers such as public perception and environmental awareness. The next barrier is infrastructure, which is resultant of several issues, including charging networks and battery recycling systems, both of which require collaboration between the private and government sectors to overcome obstacles. Finally, policy barriers have been mentioned in several previous studies.

Several previous studies have identified barriers. Haddadian et al. [27] categorized the barriers to worldwide EV adoption into four essential categories: EV technology; financial, social, and customer perception; and innovative business models. For the EV technology barrier, there are two sub-barriers: battery technology that involves high battery cost, driving range, and battery safety, and the charging station, which is an important barrier to widening EV adoption. Financial barriers focus on total cost of ownership (TCO) and financial mechanisms. Finally, innovative business models that attempt to identify perceived challenging aspects of EV holders could provide a creative approach to create and capture EV values, facilitating an easier transition to wider EV adoption. She [15] classified and examined China's [17] economic barriers that consist of rising purchase values, higher initial cost of batteries, a lack of understanding of fuel prices, and maintenance expenses. Similarly, efficiency barriers include vehicle trustworthiness, driving distance, battery recharging duration, and EV power. Facility barriers include lack of infrastructure in public locations, places of business, residences, and motorways. In order to accelerate the

EV transition, the optimal planning of electric vehicle charging station is conducted [28–31]. Adhikari et al. [16] studied the EV transition limitation using the analytic hierarchy process (AHP). The identified barriers were separated into five categories: economic, social, policy, infrastructure, and technical. Their survey revealed expert opinions. For the barrier category, the top-ranked barrier was insufficient infrastructure, followed by policy, economic, and technical barriers. The social barrier is the least concerning among the five in the list. Moreover, the study reveals that insufficient goal setting and long-term planning from the government, charging station shortage, and EVs' higher purchase price were identified as the three highest barriers to the adoption of EVs. Tarei et al. [14] mainly studied the relationship between EV barriers with one another. They also focused on the bonds between automakers and the government for strategic planning. Their methodology integrates two MCDM tools, which are the BWM best–worst method (BWM) and interpretive structural modeling (ISM). This study has five barrier categories: technical, infrastructure, financial, behavioral, and external barriers. The results showed that infrastructure barriers were highlighted as the highest based on expert evaluation, followed by financial and behavioral barriers.

## 3. Research Method

This study proposes two multicriteria decision methodologies to identify and prioritize the barriers to electric vehicle transition, as illustrated in Figure 1. The first step was a literature review that was a comprehensive review of previously published research, articles, and government publications. This process can reveal multiple aspects, difficulties, and inadequacies with regard to EVs. Moreover, national transportation policy and strategy, market trend, geography, economic status, and renewable energy resource readiness might be appropriate variables for EV barrier identification and categorization. The ANP and DEMATEL methodologies were selected because considering the nature of the problem in the real world, each barrier definitely has an influence on each other. The ANP methodology constructs a problem as a network model that is suitable for problems, and the DEMATEL methodology identifies influences through the analysis of elements in cause-and-effect relations. It has the advantages of not depending on a large data sample and simplifying the correlation analysis of factors. However, the classical DEMATEL ignores the vagueness and uncertainties of human judgement which widely exist in the real world. ANP considers the interdependencies and feedback among factors. Both selected methods use pairwise comparison to measure the weights of the components, which is reasonable for result comparisons.

### 3.1. Criterial Selection

The adoption of electric vehicles is limited by various real and perceived barriers. These barriers were revealed after a comprehensive literature review that included an investigation of relevant online articles, as well as previous research papers. An exhaustive literature search was carried out using keywords such as electric vehicles and barriers. The literature review was conducted using online search tools such as Google Scholar and the Science Direct website. Table 2 lists these barriers and their associated classes.

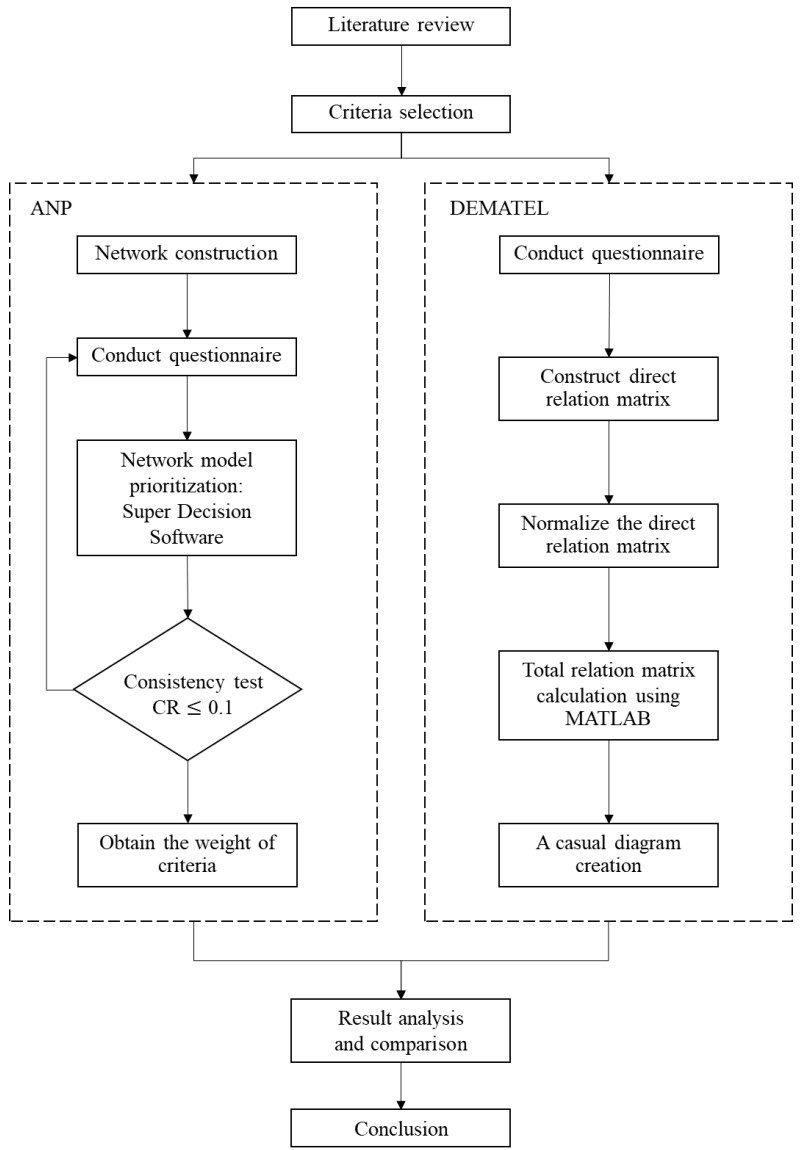

**Figure 1.** Research flowchart.

**Table 2.** Barriers of EV transition.

| Barriers | Sub Barriers | Index | References |
|---|---|---|---|
| Financial | High manufacturing cost | $F_1$ | [14,26] |
| | High selling price | $F_2$ | [32,33] |
| Infrastructure | Shortage of charging station | $I_1$ | [34,35] |
| | Lack of battery recycling system | $I_2$ | [9,26,36] |
| | Shortage of Maintenance shop | $I_3$ | [37] |
| Technology | EV Performance | $T_1$ | [14,38,39] |
| | Battery capacity and lifespan | $T_2$ | [27,36] |
| Policy | Government support | $P_1$ | [40,41] |
| | Impacts of tax and subsidy policies | $P_2$ | [42,43] |
| | Renewable energy ecosystem | $P_3$ | [40,44] |
| Customer behavior | Customer awareness | $C_1$ | [16,45] |
| | Range anxiety | $C_2$ | [46,47] |

As a result, 12 barriers to EV adoption were highlighted. Afterwards, the barriers were divided into five categories: financial, infrastructural, technological, customer behavior, and policy.

(1)　Financial:

　　(a)　High cost: High initial costs, especially for batteries, are due to the limited availability of technology and raw material issues that restrict electric vehicle adoption [14,26].

　　(b)　High prices: Conventional vehicles have a cost advantage that makes consumers more hesitant to buy EVs, which are often more expensive [32]. From the consumer perspective, price plays a major role in car purchase decisions [32,33].

(2)　Infrastructure:

　　(a)　Shortage of charging stations: Slow charging is suitable for home and work, whereas fast charging is best for places such as highways and commercial areas where cars stop for shorter periods of time. A shortage of charging stations has been identified as a barrier to consumers purchasing electric vehicles [34,35]

　　(b)　Lack of a battery recycling system: An EV's battery material consists of chemical elements such as titanium, nickel, and lithium. Although this increases the cost-effectiveness of the supply chain, it remains an environmental concern [48]. It is essential to develop biologically degradable batteries and recycling systems for them; otherwise, they might hold back the long-term sustainability of electric vehicles [9,26,36].

　　(c)　Shortage of maintenance shop: A service professional should be able to fix, maintain, and troubleshoot an EV to take appropriate care of it [48]. Current electric vehicle owners are dissatisfied with the lack of support centers or workshops for EV repair and maintenance [37].

(3)　Technology:

　　(a)　EV performance: At present, PHEVs that operate with ICEs provide driving distances of approximately 500 km. Most BEVs have a range of less than 250 km per charge. Nevertheless, some of the most recent models provide a range of up to 400 km. The driving range of a fully charged EV is regarded as a significant disadvantage in terms of EV adoption worldwide [48]. Charging duration is also a main cause for purchase consideration. A conventional vehicle is powered by fuel, which has a short duration for filling the gas tank. However, EVs remain uncertain as to how long they need to recharge the battery in case of a long journey.

　　(b)　Battery capacity and lifespan: Electric automobiles are typically constructed by substituting the fuel tank of a normal vehicle with batteries and chargers. Because EV batteries are built to last for a long period of time, they will eventually wear out at a specific time. Most suppliers currently provide an 8-year or 100,000-mile warranty [48]. Limited battery life requires frequent replacement, which is a major burden for EV users [27,36].

(4)　Customer behavior:

　　(a)　Customer awareness: Consumer awareness is essential to bringing new customers and maintaining current ones. Despite the expanding range of electric vehicles on the market, the choice of buying an electric car remains limited and is estimated to remain so in the future. As a result, automobile companies should be aware of the power of consumer offerings through marketing, social networks, or any other channels. Potential users' awareness of the benefits of EVs such as financial incentives, infrastructure availability, and potential fuel-related savings are likely to be essential factors affecting their uptake [16,45].

　　(b)　Range anxiety: Consumers' range anxiety is important in making decisions about charging station placement, and they are influenced by driving range

when they have to decide how much they are willing to drive to reach another charging station [46,47]. The driver needs to carefully organize their journey and may not be able to drive long distances.

(5) Policy:

    (a)   Government support: Government policy, city planning, and power utilities play a major role in the EV charging infrastructure and consumer awareness of the environment [40].

    (b)   Impact of tax and subsidy policies: Vehicle taxes and purchase subsidies are frequently used to provide incentives for the adoption of EVs. Compared to the consumer side, governmental tax and subsidy policies for manufacturers have a better effect on EV diffusion [42].

    (c)   Renewable energy ecosystem: However, current renewable energies are insufficient for electricity demand. Government strategy is an essential part of supporting renewable energy ecosystems by focusing on some of the most promising alternatives [40,44].

### 3.2. ANP

**Step 1. Problem structuring and network model construction**

According to the ANP procedure, weightings for pairwise comparison completion were given to each factor via a questionnaire issued to experts to collect opinions related to the relative importance of different factors. The weights of the criteria and alternatives can be evaluated from the comparison matrices that were created using a 1–9 scale.

**Step 2. Creating pairwise comparison matrices**

Pairwise comparison was completed by collecting experts' opinions through a questionnaire. Following the gathering of the experts' assessment results and preferences, we constructed a comparison matrix of clusters and criteria. Afterward, a review of the consistency ratio (*C.R.*) of the matrices was required. The pairwise comparison is considered acceptable when it is less than 0.1 for every comparison matrix. In the case that $C.I. > 0.1$, it means that decision-makers and experts have differing viewpoints. It is necessary to alter their assessments to create a new and consistent comparison matrix [49]. The consistency ratio (*C.R.*) is defined as follows:

$$C.R. = \frac{C.I.}{R.I.}$$

where $C.I. = \frac{\lambda_{max} - n}{n - 1}$, represents the consistency index and *R.I.* represents random index. Table 3 shows the value of *R.I.*

**Table 3.** The value of *R.I.*

| | 1 | 2 | 3 | 4 | 5 | 6 | 7 | 8 | 9 |
|---|---|---|---|---|---|---|---|---|---|
| *R.I.* | 0 | 0 | 0.58 | 0.9 | 1.12 | 1.24 | 1.32 | 1.41 | 1.45 |

**Step 3. Super-matrix formation and ranking the criteria**

Following the completion of the pairwise matrix comparison, a super-matrix was built using prioritized vectors derived from the previous step for interdependency. A normalizing process was constructed to build a weighted super-matrix. Then, for the limited matrix, a raw value of the weighted matrix equal to each column of the super-matrix using the vectors calculated from the previous step to conduct the super-matrix columns was made, where each segment of the partitioned matrix denotes a relationship between two clusters. A considered network is broken down into $N$ clusters, displayed by $C_1, C_2, \ldots, C_N$ and the elements in $Ck, 1 \leq k \leq N$ are $e_{k1}, e_{k2}, \ldots, e_{knk}$ where $n_k$ represents the number of elements in the $Ck$ cluster. These processes were carried out using Super Decisions, an ANP-developed software tool. The synthesis process determines the overall

priority of each alternative. The results of each subnetwork are merged to assign the final priorities of the alternatives, which are then ranked.

### 3.3. DEMATEL

Step 1. Conduct questionnaire

A questionnaire with previously identified criteria and collaboration with experts was conducted to complete the direct relation matrix. The respondents were requested to identify the influence of each criterion on others using $nxn$ pairwise comparisons. The questionnaire used a numerical score that ranged from 0 to 4. The score representative sequent is shown as follows: 0, no influence; 1, low influence; 2, medium influence; 3, high influence; and 4, very high influence.

Step 2. Construct direct relation matrix

Assume that, for this study, there were $H$ experts and $n$ factors for consideration. All respondents were required to indicate the level at which they thought a factor, say i, influences another factor j. $X_{ij}^K$ represents pairwise comparisons of the opinion provided by $k$th experts about the $i$th and $j$th factors which received a score in the previous step. The score given by the respondent was assigned to the $n \times n$ nonnegative response matrix $X^k = [x_{ij}^k]$ with $k = 1, 2, \ldots, H$. Therefore, for each $H$ expert, the answer matrices were $X^1$, $X^2, \ldots, X^H$. Moreover, each component of $X^k = [x_{ij}^k]$ is an integer indicated by $X_{ij}^K$. Each $X^k = [x_{ij}^k]$ response was diagonal, while the other elements were defined as zero. Then, it was possible to calculate $n \times n$ average matrix $A$ for all respondent evaluations by equalizing the given degree by $H$ respondents as follows:

$$[a_{ij}]_{nxn} = \frac{1}{H} \sum_{k=1}^{H} [x_{ij}^k]_{nxn}$$

The average direct relation matrix $A = [x_{ij}^k]_{nxn}$ is also called the initial direct relation. Matrix A illustrates the factor's initial direct effects on and obtained from the other factors. Moreover, an influence map was drawn to visualize the causal effect of each aspect of the system. Each node in the map represents a system factor, with arrows corresponding to the interactions between them. As an example, an arrow from C2 to C4 indicates the impact that C2 has over C4, and the effect strength is 1. Using DEMATEL, an intelligible map of the system was created from the structural relations among the factors in the system.

Step 3. Normalize the direct relation matrix

The normalized initial direct-relation matrix $D = [d_{ij}]_{nxn}$ was acquired by applying the following formula to normalize the average matrix A:

$$S = max\left\{ \max_{1 \le i \le n} \sum_{j=1}^{n} aij, \max_{1 \le j \le n} \sum_{i=1}^{n} aij \right\}, D = \frac{A}{S}$$

Consequently, the direct effects of each factor on other factors are represented by the total of each row $j$ of matrix $A$, and $\max_{1 \le i \le n} \sum_{j=1}^{n} aij$ indicates the factor with the largest direct influence on the other factors. The direct effects received by factor $i$ are represented by the sum of each column $i$ of matrix $A$, and $\max_{1 \le j \le n} \sum_{i=1}^{n} aij$ indicates the factor that is most influenced by other factors. The larger of the two extreme sums is equal to the positive scalar $s$. By dividing each element of A by the scalar s, the matrix $D$ was constructed. Each $d_{ij}$ element in matrix $D$ had a value between 0 and 1.

Step 4. Total relation matrix calculation using MATLAB

The power of the normalized initial direct relation matrix $D$ is represented by $D^m$, which is an m-indirect effect that can be used to illustrate the length effect, or the influence propagated by m-1 intermediates. Convergent solutions to the matrix inversion are guaranteed by a continuous reduction in the indirect effects of problems other than the powers of matrix $D$, such as an engrossing Markov chain matrix. The total influence or relationship can be obtained by summing up $D^1, D^2, D^3, \ldots, D^\infty$,

$$\lim_{m \to \infty} D^m = [0]_{nxn}, \tag{1}$$

where $[0]_{nxn}$ is a $nxn$ null matrix.

The total relation matrix $T_{nxn}$ is accomplished as follows:

$$\sum_{m=1}^{\infty} D_i = D + D^2 + D^3 \ldots D^M$$
$$= D(I + D + D^2 + \ldots + D^{M-1})$$
$$= D(I - D)^{-1}(I - D)(I + D + D^2 + \ldots + D^{M-1})$$
$$= D(I - D)^{-1}(I - D^m) = D(I - D)^{-1}$$

$T$ is total relation matrix ($[T]_{nxn}$). $I$ is identity matrix.

The total relation matrix's sum of rows and columns is generated as $r$ and $c$ $nx1$ vectors.

$$r = [r_i]_{nx1} = \left(\sum_{j=1}^{n} t_{ij}\right)_{nx1} \quad, \quad c = [c_j]_{1xn} = \left(\sum_{i=1}^{n} t_{ij}\right)_{1xn}$$

$[r_i]_{nx1}$ is the sum of the $i$th row of matrix $T$ and identifies the overall effect, including indirect and direct, indicated to other factors by factor $I$. Additionally, the sum of the $j$th column of matrix $T$ is $[c_j]_{1xn}$ and both the indirect and direct of the total effects obtained by factor $j$ from other factors can be denoted by $[c_j]_{1xn}$. Furthermore, the total effects, including those received and given by $i$-th factor, can be represented by the sum $(r_i + c_i)$. Otherwise, the value of $(r_i + c_i)$ indicates the importance of the $i$-th factor in the system (received and given effects in total). Moreover, the net effect that factor $i$ contributes to the system can be represented by the difference (the relation which is $r_i - c_i$). Factor $i$ is a net causer if $(r_i - c_i)$ is positive. On the other hand, factor $i$ is a net receiver if $(r_i - c_i)$ is negative [50,51].

## 4. Results and Discussion

### 4.1. ANP Priority Analysis

The network model was built by considering the relationship between the real world and existing research. After the criteria selection was complete, Super Decision software was used to execute the pairwise comparison process. Table 4 presents the unweighted super-matrix of this study. Then the weighted super-matrix and limited super-matrix show as Tables A1 and A2 in Appendix B, respectively. Table 5 shows the weight of EV barriers.

### 4.2. DEMATEL Priority Analysis

Through the literature review, 12 criteria for EV barriers were identified. A $12 \times 12$ pairwise comparison questionnaire was administered to analyze the interrelationships among criteria. After collecting all questionnaires from the experts, the relationships between the criteria were assessed using a 0–4 scale. An average direct relation matrix was constructed, as shown in Tables A3–A5 in Appendix B. For this step, the matrix was computed by using MATLAB software.

**Table 4.** Unweighted super-matrix.

| Barrier | Sub-Barrier | Financial | | Infrastructure | | | Technology | | Customer Behavior | | Policy | | |
|---|---|---|---|---|---|---|---|---|---|---|---|---|---|
| | | F1 | F2 | I1 | I2 | I3 | T1 | T2 | C1 | C2 | P1 | P2 | P3 |
| Financial | F1 | 0.000 | 1.000 | 0.857 | 0.000 | 0.875 | 0.857 | 0.857 | 0.833 | 0.000 | 0.857 | 0.857 | 0.000 |
| | F2 | 1.000 | 0.000 | 0.143 | 1.000 | 0.125 | 0.143 | 0.143 | 0.167 | 0.000 | 0.143 | 0.143 | 0.000 |
| Infrastructure | I1 | 0.571 | 0.000 | 0.000 | 1.000 | 0.000 | 0.000 | 1.000 | 0.750 | 0.750 | 1.000 | 0.000 | 0.000 |
| | I2 | 0.143 | 0.000 | 0.000 | 0.000 | 0.000 | 0.000 | 0.000 | 0.000 | 0.000 | 0.000 | 0.000 | 0.000 |
| | I3 | 0.286 | 0.000 | 0.000 | 0.000 | 0.000 | 0.000 | 0.000 | 0.250 | 0.250 | 0.000 | 0.000 | 0.000 |
| Technology | T1 | 0.111 | 0.000 | 0.000 | 0.000 | 0.000 | 0.111 | 1.000 | 0.111 | 0.111 | 0.000 | 0.111 | 0.000 |
| | T2 | 0.889 | 0.000 | 1.000 | 0.000 | 0.000 | 0.889 | 0.000 | 0.889 | 0.889 | 0.000 | 0.889 | 0.000 |
| Customer behavior | C1 | 0.000 | 1.000 | 0.889 | 0.000 | 0.889 | 0.000 | 0.000 | 0.000 | 1.000 | 1.000 | 0.000 | 1.000 |
| | C2 | 0.000 | 0.000 | 0.111 | 0.000 | 0.111 | 1.000 | 0.000 | 1.000 | 0.000 | 0.000 | 0.000 | 0.000 |
| Policy | P1 | 0.667 | 0.667 | 0.667 | 0.000 | 0.000 | 0.667 | 0.667 | 0.588 | 0.000 | 0.000 | 1.000 | 0.000 |
| | P2 | 0.333 | 0.333 | 0.333 | 0.000 | 0.000 | 0.333 | 0.333 | 0.323 | 0.000 | 1.000 | 0.000 | 0.000 |
| | P3 | 0.000 | 0.000 | 0.000 | 0.000 | 0.000 | 0.000 | 0.000 | 0.089 | 0.000 | 0.000 | 0.000 | 0.000 |

**Table 5.** Weights of EV barriers.

| Barrier | Weight | Sub-Barrier | Normalized by Cluster | Limiting | Rank |
|---|---|---|---|---|---|
| Financial | 0.194 | High cost (F1) | 0.762 | 0.148 | 3 |
| | | High price (F2) | 0.238 | 0.046 | 7 |
| Infrastructure | 0.072 | Shortage of charging stations (I1) | 0.898 | 0.065 | 6 |
| | | Lack of battery recycling system (I2) | 0.030 | 0.002 | 11 |
| | | Shortage of maintenance shops (I3) | 0.072 | 0.005 | 10 |
| Technology | 0.365 | EV performance (T1) | 0.356 | 0.130 | 5 |
| | | Battery capacity and lifespan (T2) | 0.644 | 0.235 | 1 |
| Customer behavior | 0.049 | Customer awareness (C1) | 0.728 | 0.036 | 8 |
| | | Range anxiety (C2) | 0.272 | 0.013 | 9 |
| Policy | 0.319 | Government support (P1) | 0.548 | 0.175 | 2 |
| | | Impacts of tax and subsidy policies (P2) | 0.449 | 0.143 | 4 |
| | | Renewable energy ecosystem (P3) | 0.003 | 0.001 | 12 |

Using the matrix results, the final result of DEMATEL analysis using MATLAB software was calculated. The mean value for all criteria was computed as the threshold value, which was 0.4116. The value of $D + R$ and $D - R$ are calculated and illustrated in Table 6 and Figure 2. The $D + R$ stands for the degree of central role that the factor plays in the system. Similarly, the vertical axis vector called "Relation" shows the net effect that the factor contributes to the system. In addition, the $D + R$ and $D - R$ could be identified as cause factors of perceived benefits, effect factors of perceived benefits, cause factors of perceive risks, and effect factors of perceived risks. In addition, a threshold value is set in many studies to filter out negligible effects. That is, only the elements of the matrix whose influence levels are greater than the value of are selected and converted. If the threshold value is too low, too many factors are included and the IRM is too complex to comprehend. In contrast, some important factors may be excluded if the threshold value is too high. In the literature, the threshold value is usually determined by experts through discussions.

**Table 6.** Final results of DEMATEL analysis.

| No. | Criteria | D | R | D−R | D+R |
|---|---|---|---|---|---|
| 1 | High cost (F1) | 5.438 | 5.185 | 0.252 | 10.623 |
| 2 | High price (F2) | 5.049 | 5.433 | −0.384 | 10.482 |
| 3 | Shortage of charging stations (I1) | 5.131 | 4.891 | 0.240 | 10.021 |
| 4 | Lack of battery recycling system (I2) | 3.597 | 4.333 | −0.736 | 7.930 |
| 5 | Shortage of maintenance shops (I3) | 4.128 | 4.337 | −0.209 | 8.466 |
| 6 | EV performance (T1) | 5.236 | 4.827 | 0.409 | 10.063 |
| 7 | Battery capacity and lifespan (T2) | 5.938 | 5.007 | 0.930 | 10.945 |
| 8 | Customer awareness (C1) | 4.880 | 5.713 | −0.833 | 10.593 |
| 9 | Range anxiety (C2) | 5.064 | 4.377 | 0.687 | 9.440 |
| 10 | Government support (P1) | 5.685 | 5.241 | 0.445 | 10.926 |
| 11 | Impacts of tax and subsidy policies (P2) | 5.500 | 5.419 | 0.081 | 10.919 |
| 12 | Renewable energy ecosystem (P3) | 3.624 | 4.506 | −0.882 | 8.130 |

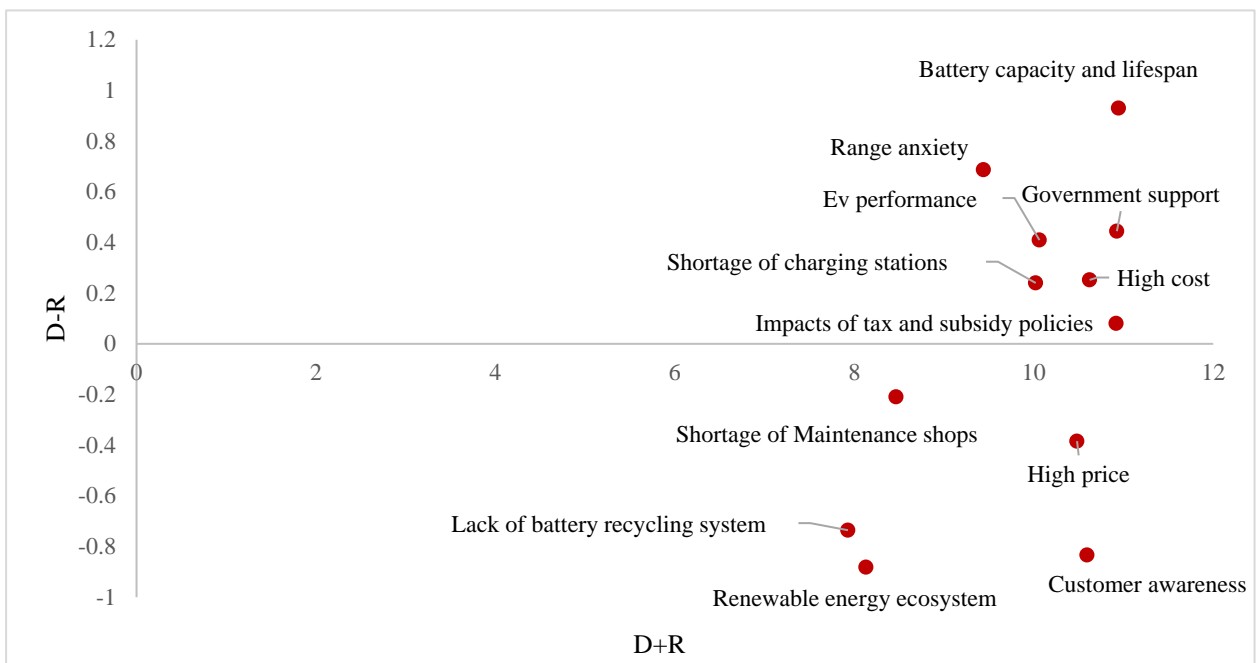

**Figure 2.** The casual diagram.

*4.3. Result Analysis*

From the 12 selected barriers to EV transition, this study compared the application of Ctwo different multicriteria decision methodologies for the analysis. Table 7 displays the summary results of the weighting of EV transition barriers using the ANP and DEMATEL methods.

The results illustrate that "battery capacity and lifespan" from the EV technology category is the most important barrier to EV transition from both methodologies, which implies that the largest obstacle to EV adoption for the automotive sector is battery technology development. Next, the second most important barrier from ANP and DEMATEL methodology is "government support" from the policy category which includes long-term policy, city planning, power utilities etc.

**Table 7.** Result comparison.

| Rank | ANP | DEMATEL |
|:---:|---|---|
| 1 | Battery capacity and lifespan | Battery capacity and lifespan |
| 2 | Government support | Government support |
| 3 | High cost | Impacts of tax and subsidy policies |
| 4 | Impacts of tax and subsidy policies | High cost |
| 5 | EV performance | Customer awareness |
| 6 | Shortage of charging stations | High price |
| 7 | High price | EV performance |
| 8 | Customer awareness | Shortage of charging stations |
| 9 | Range anxiety | Range anxiety |
| 10 | Shortage of maintenance shops | Shortage of maintenance shops |
| 11 | Lack of battery recycling system | Renewable energy ecosystem |
| 12 | Renewable energy ecosystem | Lack of battery recycling system |

When the sequence between the fifth place and the eighth place of the summary result is compared, it can be seen that the importance sequence of each barrier slightly fluctuates between "EV performance", "shortage of charging stations", "high price", and "customer awareness". Table 7 shows that "lack of battery recycling system", "renewable energy ecosystem", and "shortage of maintenance shops" are the final important barriers to EV transition. However, the priority of the last two barriers, "lack of battery recycling system" and "renewable energy ecosystem" are swapped because of the different stages of the utilized methods.

*4.4. Summary of the Results*

The result comparison of the two multicriteria decision-making (MCDM) methods were applied in this study. The summarized outputs are presented. Among the 5 barrier categories and 12 sub-barriers analyzed, the EV technology barrier category was prioritized, after first referring to the opinions and analyses of participating experts. Within the major category, battery capacity and lifespan play a pivotal role among the sub-barriers, and not only has the highest weighting but also the highest influence on other barriers according to the DEMATEL result.

Notably, the policy barrier category was the second-highest-ranked barrier in this study. Within the policy category, the government support barrier had the highest weighting and second most important rank among all sub-barriers. This output strengthens the research findings, which highlight that government reinforcement is a critical determinant of consumers' adoption of EVs. Additionally, a study in China also emphasizes the government's role in various aspects that support the achievement of EVs, such as R&D investment, EV demonstration programs, and electric vehicle promotion strategies [52]. Moreover, another sub-barrier listed in the policy category and ranked in the top four of the overall result is the impact of tax and subsidy policy barriers. This sub-barrier has been mentioned and investigated in several previous studies. For example, research in Europe [42] concluded that tax and subsidy impacts at various points include increasing EV sales by reducing total ownership costs and increasing environmental benefits by implementing a green tax. Another study [53] also highlighted the importance of subsidy policy on EV adoption; the subsidy policy from the public sector increases EV cost compatibility with ICE vehicles.

From a manufacturer's perspective, a shortage of charging stations is not very important. It is completely different from the consumer perspective in that the shortage of charging stations is the main concern of potential EV purchasers. In addition, this study surveyed most data in Thailand. When compared with related topics in Taiwan [54], the direction of EV policy is considerably influenced by the leading manufacturer, which differs from the results of this study, indicating the importance of policy over the enterprise. Moreover, the results from the two different methods were similar owing to the consistent estimation of experts. It is worth mentioning, although in this study, only the emission

values of the vehicles at the time of use were considered. However, the effect of these vehicles on GHG during the production phase should also be investigated.

## 5. Conclusions

The near-term future of EV sales is high. In the first quarter of 2021, global electric car sales rose by approximately 140% compared to the same period in 2020 [3]. To summarize, as mentioned in the previous section, previous studies were considered and analyzed to identify barriers to EV transition. After identifying barriers related to EV diffusion, the ANP and DEMATEL methodologies were applied.

The first research output was a list of 12 barriers to EV transition under five categories identified from a comprehensive review, including financial, infrastructure, technology, customer behavior, and policy. The second output research was the prioritized result from the analytic network process (ANP) and the decision-making trial and evaluation laboratory (DEMATEL) technique performed concurrently. The highest weighted barrier was battery capacity and lifespan, followed by government support, the impacts of tax and subsidy policies, and high costs. Moreover, the outcome from the DEMATEL methodology showed that high weighing barriers such as the battery capacity, lifespan, and government support barriers have influence on other barriers such as high price barriers, customer awareness barriers, and lack of battery recycling systems.

Managerial implications for the EV industry: This study's findings emphasize the urgent barrier to electric vehicle transition for the automotive sector and improve the understanding of private and public sector corporations. Battery capacity and lifespan require major concentration and enhancement from manufacturers, which greatly influence electric vehicle performance, range anxiety, and awareness of EV customers. Moreover, government support plays a key role. The scarcity of sales offices and charging time are both also barriers. Policy makers need to pay attention to EVs supporting regulations such as driving up the number of public charging stations, decreasing electricity fees, and encouraging renewable energy enterprises. Eventually, the successful adoption of electric vehicles will result in lower consumption of fossil fuels, natural resource usage optimization, and sustainability.

The main contributions of this study are as follows: First, the research identified and prioritized the barriers that manufacturers confront in making the transition to electric vehicles, as well as the interrelationships between these barriers. Second, this study proposed a comparison model of two multi criteria decision methodologies for prioritizing and identifying the interrelation of the EV adoption barrier, which potentially provides more robust results than applying only one multi criteria decision methodology.

A limitation of this study is that the methodology input greatly depends on the judgement and estimation of experts. However, the number of respondents was limited because of the high level of experience required by the respondents.

In future research, other barriers against EV transition that may spring up in the years ahead could be analyzed. To define such challenges consistently, regular and continuous literature reviews and interrelations with consumers, automakers, specialists, and policymakers are necessary. This study can be applied to EV barrier analysis. Therefore, model applicability should be reviewed before it can be used in a specific context. Different MCDM tool combinations can be used to examine the robustness of the study findings.

**Author Contributions:** Conceptualization, T.-C.K. and N.S.; methodology and writing, T.-C.K. and N.S.; validation, Y.-S.S.; formal analysis, N.S.; investigation, T.-C.K., N.S. and R.-H.Y. All authors have read and agreed to the published version of the manuscript.

**Funding:** This research was funded by the National Science and Technology Council of the Republic of China, Taiwan, grant number 110-2621-M-011-002-MY3 and NTUST- Kind-9399.

**Conflicts of Interest:** The authors declare no conflict of interest.

## Appendix A. Survey

Instruction: Please rate each question in the horizontal row based on your experience using the following scale

| Importance Scale | Definition of Importance Scale |
|---|---|
| 1 | Equal importance preferred |
| 2 | Equal to moderate importance preferred |
| 3 | Moderate importance preferred |
| 4 | Moderate to strong importance preferred |
| 5 | Strong importance preferred |
| 6 | Strong to very strong importance preferred |
| 7 | Very strong importance preferred |
| 8 | Very strong to extreme importance preferred |
| 9 | Extreme importance preferred |

0. With respect to infrastructure, which is more important to electric vehicle transition barriers between:

| No | Barrier | More Importance on the Left | | | | | | | | | More Importance on the Right | | | | | | | | Barrier |
|---|---|---|---|---|---|---|---|---|---|---|---|---|---|---|---|---|---|---|---|
| | | 9 | 8 | 7 | 6 | 5 | 4 | 3 | 2 | 1 | 2 | 3 | 4 | 5 | 6 | 7 | 8 | 9 | |
| 0 | Customer behavior | | | | | | | | | | | | | √ | | | | | Financial |

With respect to Infrastructure, if you think "Financial" is strongly importance to electric vehicle transition barrier more than "Customer behavior" then check 5 on the right

1. With respect to finance, which is more important to electric vehicle transition barrier between

| No | Barrier | More Importance on the Left | | | | | | | | | More Importance on the Right | | | | | | | | Barrier |
|---|---|---|---|---|---|---|---|---|---|---|---|---|---|---|---|---|---|---|---|
| | | 9 | 8 | 7 | 6 | 5 | 4 | 3 | 2 | 1 | 2 | 3 | 4 | 5 | 6 | 7 | 8 | 9 | |
| 1 | Customer behavior | | | | | | | | | | | | | | | | | | Financial |
| 2 | Customer behavior | | | | | | | | | | | | | | | | | | Infrastructure |
| 3 | Customer behavior | | | | | | | | | | | | | | | | | | Policy |
| 4 | Customer behavior | | | | | | | | | | | | | | | | | | Technology |
| 5 | Financial | | | | | | | | | | | | | | | | | | Infrastructure |
| 6 | Financial | | | | | | | | | | | | | | | | | | Policy |
| 7 | Financial | | | | | | | | | | | | | | | | | | Technology |
| 8 | Infrastructure | | | | | | | | | | | | | | | | | | Policy |
| 9 | Infrastructure | | | | | | | | | | | | | | | | | | Technology |
| 10 | Policy | | | | | | | | | | | | | | | | | | Technology |

2. With respect to infrastructure, which is more important to electric vehicle transition barriers between:

| No | Barrier | More Importance on the Left | | | | | | | | More Importance on the Right | | | | | | | | Barrier |
|---|---|---|---|---|---|---|---|---|---|---|---|---|---|---|---|---|---|---|
| | | 9 | 8 | 7 | 6 | 5 | 4 | 3 | 2 | 1 | 2 | 3 | 4 | 5 | 6 | 7 | 8 | 9 | |
| 1 | Customer behavior | | | | | | | | | | | | | | | | | | Financial |
| 2 | Customer behavior | | | | | | | | | | | | | | | | | | Infrastructure |
| 3 | Customer behavior | | | | | | | | | | | | | | | | | | Policy |
| 4 | Customer behavior | | | | | | | | | | | | | | | | | | Technology |
| 5 | Financial | | | | | | | | | | | | | | | | | | Infrastructure |
| 6 | Financial | | | | | | | | | | | | | | | | | | Policy |
| 7 | Financial | | | | | | | | | | | | | | | | | | Technology |
| 8 | Infrastructure | | | | | | | | | | | | | | | | | | Policy |
| 9 | Infrastructure | | | | | | | | | | | | | | | | | | Technology |
| 10 | Policy | | | | | | | | | | | | | | | | | | Technology |

3. With respect to technology, which is more important to electric vehicle transition barriers between:

| No | Barrier | More Importance on the Left | | | | | | | | More Importance on the Right | | | | | | | | Barrier |
|---|---|---|---|---|---|---|---|---|---|---|---|---|---|---|---|---|---|---|
| | | 9 | 8 | 7 | 6 | 5 | 4 | 3 | 2 | 1 | 2 | 3 | 4 | 5 | 6 | 7 | 8 | 9 | |
| 1 | Customer behavior | | | | | | | | | | | | | | | | | | Financial |
| 2 | Customer behavior | | | | | | | | | | | | | | | | | | Infrastructure |
| 3 | Customer behavior | | | | | | | | | | | | | | | | | | Policy |
| 4 | Customer behavior | | | | | | | | | | | | | | | | | | Technology |
| 5 | Financial | | | | | | | | | | | | | | | | | | Infrastructure |
| 6 | Financial | | | | | | | | | | | | | | | | | | Policy |
| 7 | Financial | | | | | | | | | | | | | | | | | | Technology |
| 8 | Infrastructure | | | | | | | | | | | | | | | | | | Policy |
| 9 | Infrastructure | | | | | | | | | | | | | | | | | | Technology |
| 10 | Policy | | | | | | | | | | | | | | | | | | Technology |

4. With respect to customer behavior, which is more important to electric vehicle transition barriers between:

| No | Barrier | More Importance on the Left | | | | | | | | More Importance on the Right | | | | | | | | Barrier |
|---|---|---|---|---|---|---|---|---|---|---|---|---|---|---|---|---|---|---|
| | | 9 | 8 | 7 | 6 | 5 | 4 | 3 | 2 | 1 | 2 | 3 | 4 | 5 | 6 | 7 | 8 | 9 | |
| 1 | Customer behavior | | | | | | | | | | | | | | | | | | Financial |
| 2 | Customer behavior | | | | | | | | | | | | | | | | | | Infrastructure |
| 3 | Customer behavior | | | | | | | | | | | | | | | | | | Policy |
| 4 | Customer behavior | | | | | | | | | | | | | | | | | | Technology |
| 5 | Financial | | | | | | | | | | | | | | | | | | Infrastructure |
| 6 | Financial | | | | | | | | | | | | | | | | | | Policy |
| 7 | Financial | | | | | | | | | | | | | | | | | | Technology |
| 8 | Infrastructure | | | | | | | | | | | | | | | | | | Policy |
| 9 | Infrastructure | | | | | | | | | | | | | | | | | | Technology |
| 10 | Policy | | | | | | | | | | | | | | | | | | Technology |

5. With respect to policy, which is more important to electric vehicle transition barriers between:

| No | Barrier | More Importance on the Left | | | | | | | | More Importance on the Right | | | | | | | | Barrier |
|----|---------|---|---|---|---|---|---|---|---|---|---|---|---|---|---|---|---|---------|
| | | 9 | 8 | 7 | 6 | 5 | 4 | 3 | 2 | 1 | 2 | 3 | 4 | 5 | 6 | 7 | 8 | 9 | |
| 1 | Customer behavior | | | | | | | | | | | | | | | | | | Financial |
| 2 | Customer behavior | | | | | | | | | | | | | | | | | | Infrastructure |
| 3 | Customer behavior | | | | | | | | | | | | | | | | | | Policy |
| 4 | Customer behavior | | | | | | | | | | | | | | | | | | Technology |
| 5 | Financial | | | | | | | | | | | | | | | | | | Infrastructure |
| 6 | Financial | | | | | | | | | | | | | | | | | | Policy |
| 7 | Financial | | | | | | | | | | | | | | | | | | Technology |
| 8 | Infrastructure | | | | | | | | | | | | | | | | | | Policy |
| 9 | Infrastructure | | | | | | | | | | | | | | | | | | Technology |
| 10 | Policy | | | | | | | | | | | | | | | | | | Technology |

6. With respect to high cost, which is more important to electric vehicle transition barriers between:

| No | Barrier | More Importance on the Left | | | | | | | | More Importance on the Right | | | | | | | | Barrier |
|----|---------|---|---|---|---|---|---|---|---|---|---|---|---|---|---|---|---|---------|
| | | 9 | 8 | 7 | 6 | 5 | 4 | 3 | 2 | 1 | 2 | 3 | 4 | 5 | 6 | 7 | 8 | 9 | |
| 1 | Shortage of charging stations | | | | | | | | | | | | | | | | | | Lack of battery recycling system |
| 2 | Shortage of charging stations | | | | | | | | | | | | | | | | | | Shortage of maintenance shops |
| 3 | Lack of battery recycling system | | | | | | | | | | | | | | | | | | Shortage of maintenance shops |
| 4 | EV performance | | | | | | | | | | | | | | | | | | Battery capacity and life span |
| 5 | Government support | | | | | | | | | | | | | | | | | | Impact of tax and subsidy policies |

7. With respect to high price, which is more important to electric vehicle transition barriers between:

| No | Barrier | More Importance on the Left | | | | | | | | More Importance on the Right | | | | | | | | Barrier |
|----|---------|---|---|---|---|---|---|---|---|---|---|---|---|---|---|---|---|---------|
| | | 9 | 8 | 7 | 6 | 5 | 4 | 3 | 2 | 1 | 2 | 3 | 4 | 5 | 6 | 7 | 8 | 9 | |
| 1 | Government support | | | | | | | | | | | | | | | | | | Impact of tax and subsidy policies |

8. With respect to shortage of charging stations, which is more important to electric vehicle transition barriers between:

| No | Barrier | More Importance on the Left | | | | | | | | More Importance on the Right | | | | | | | | Barrier |
|----|---------|---|---|---|---|---|---|---|---|---|---|---|---|---|---|---|---|---------|
| | | 9 | 8 | 7 | 6 | 5 | 4 | 3 | 2 | 1 | 2 | 3 | 4 | 5 | 6 | 7 | 8 | 9 | |
| 1 | High cost | | | | | | | | | | | | | | | | | | High price |
| 2 | Customer awareness | | | | | | | | | | | | | | | | | | Range anxiety |
| 3 | Government support | | | | | | | | | | | | | | | | | | Impact of tax and subsidy policies |

9. With respect to shortage of maintenance shops, which is of more importance to electric vehicle transition barriers between:

| No | Barrier | More Importance on the Left | | | | | | | | More Importance on the Right | | | | | | | | Barrier |
|---|---|---|---|---|---|---|---|---|---|---|---|---|---|---|---|---|---|---|
| | | 9 | 8 | 7 | 6 | 5 | 4 | 3 | 2 | 1 | 2 | 3 | 4 | 5 | 6 | 7 | 8 | 9 | |
| 1 | High cost | | | | | | | | | | | | | | | | | | High price |
| 2 | Customer awareness | | | | | | | | | | | | | | | | | | Range anxiety |

10. With respect to EV performance, which is of more importance to electric vehicle transition barriers between:

| No | Barrier | More Importance on the Left | | | | | | | | More Importance on the Right | | | | | | | | Barrier |
|---|---|---|---|---|---|---|---|---|---|---|---|---|---|---|---|---|---|---|
| | | 9 | 8 | 7 | 6 | 5 | 4 | 3 | 2 | 1 | 2 | 3 | 4 | 5 | 6 | 7 | 8 | 9 | |
| 1 | High cost | | | | | | | | | | | | | | | | | | High price |
| 2 | EV performance | | | | | | | | | | | | | | | | | | Battery capacity and life span |
| 3 | Government support | | | | | | | | | | | | | | | | | | Impact of tax and subsidy policies |

11. With respect to battery capacity and life span, which is of more importance to electric vehicle transition barriers between:

| No | Barrier | More Importance on the Left | | | | | | | | More Importance on the Right | | | | | | | | Barrier |
|---|---|---|---|---|---|---|---|---|---|---|---|---|---|---|---|---|---|---|
| | | 9 | 8 | 7 | 6 | 5 | 4 | 3 | 2 | 1 | 2 | 3 | 4 | 5 | 6 | 7 | 8 | 9 | |
| 1 | High cost | | | | | | | | | | | | | | | | | | High price |
| 2 | Government support | | | | | | | | | | | | | | | | | | Impact of tax and subsidy policies |

12. With respect to customer awareness, which is of more importance to electric vehicle transition barriers between:

| No | Barrier | More Importance on the Left | | | | | | | | More Importance on the Right | | | | | | | | Barrier |
|---|---|---|---|---|---|---|---|---|---|---|---|---|---|---|---|---|---|---|
| | | 9 | 8 | 7 | 6 | 5 | 4 | 3 | 2 | 1 | 2 | 3 | 4 | 5 | 6 | 7 | 8 | 9 | |
| 1 | High cost | | | | | | | | | | | | | | | | | | High price |
| 2 | EV performance | | | | | | | | | | | | | | | | | | Battery capacity and life span |
| 3 | Shortage of charging stations | | | | | | | | | | | | | | | | | | Shortage of maintenance shops |
| 4 | Government support | | | | | | | | | | | | | | | | | | Impact of tax and subsidy policies |

13. With respect to range anxiety, which is of more importance to electric vehicle transition barriers between:

| No | Barrier | More Importance on the Left | | | | | | | | More Importance on the Right | | | | | | | | Barrier |
|---|---|---|---|---|---|---|---|---|---|---|---|---|---|---|---|---|---|---|
| | | 9 | 8 | 7 | 6 | 5 | 4 | 3 | 2 | 1 | 2 | 3 | 4 | 5 | 6 | 7 | 8 | 9 | |
| 2 | EV performance | | | | | | | | | | | | | | | | | | Battery capacity and life span |
| 3 | Shortage of charging stations | | | | | | | | | | | | | | | | | | Shortage of maintenance shops |

| No | Barrier | More Importance on the Left | | | | | | | | | More Importance on the Right | | | | | | | | | Barrier |
|---|---|---|---|---|---|---|---|---|---|---|---|---|---|---|---|---|---|---|---|---|
| | | 9 | 8 | 7 | 6 | 5 | 4 | 3 | 2 | 1 | 2 | 3 | 4 | 5 | 6 | 7 | 8 | 9 | |

14. With respect to government support, which is of more importance to electric vehicle transition barriers between:

| No | Barrier | More Importance on the Left | | | | | | | | | More Importance on the Right | | | | | | | | | Barrier |
|---|---|---|---|---|---|---|---|---|---|---|---|---|---|---|---|---|---|---|---|---|
| | | 9 | 8 | 7 | 6 | 5 | 4 | 3 | 2 | 1 | 2 | 3 | 4 | 5 | 6 | 7 | 8 | 9 | |
| 1 | High cost | | | | | | | | | | | | | | | | | | High price |

15. With respect to impacts of tax and subsidy policies, which is of more importance to electric vehicle transition barriers between:

| No | Barrier | More Importance on the Left | | | | | | | | | More Importance on the Right | | | | | | | | | Barrier |
|---|---|---|---|---|---|---|---|---|---|---|---|---|---|---|---|---|---|---|---|---|
| | | 9 | 8 | 7 | 6 | 5 | 4 | 3 | 2 | 1 | 2 | 3 | 4 | 5 | 6 | 7 | 8 | 9 | |
| 1 | High cost | | | | | | | | | | | | | | | | | | High price |
| 2 | EV performance | | | | | | | | | | | | | | | | | | Battery capacity and life span |

## Appendix B

**Table A1.** Weighted super-matrix.

| Barrier | Sub-Barrier | Financial | | Infrastructure | | | Technology | | Customer Behavior | | Policy | | |
|---|---|---|---|---|---|---|---|---|---|---|---|---|---|
| | | F1 | F2 | I1 | I2 | I3 | T1 | T2 | C1 | C2 | P1 | P2 | P3 |
| Financial | F1 | 0.000 | 0.302 | 0.135 | 0.000 | 0.570 | 0.164 | 0.159 | 0.110 | 0.000 | 0.219 | 0.149 | 0.000 |
| | F2 | 0.149 | 0.000 | 0.023 | 0.850 | 0.081 | 0.027 | 0.027 | 0.022 | 0.000 | 0.037 | 0.025 | 0.000 |
| Infrastructure | I1 | 0.058 | 0.000 | 0.000 | 0.150 | 0.000 | 0.000 | 0.095 | 0.045 | 0.075 | 0.176 | 0.000 | 0.000 |
| | I2 | 0.015 | 0.000 | 0.000 | 0.000 | 0.000 | 0.000 | 0.000 | 0.000 | 0.000 | 0.000 | 0.000 | 0.000 |
| | I3 | 0.029 | 0.000 | 0.000 | 0.000 | 0.000 | 0.000 | 0.000 | 0.015 | 0.025 | 0.000 | 0.000 | 0.000 |
| Technology | T1 | 0.050 | 0.000 | 0.000 | 0.000 | 0.000 | 0.051 | 0.445 | 0.050 | 0.083 | 0.000 | 0.059 | 0.000 |
| | T2 | 0.400 | 0.000 | 0.499 | 0.000 | 0.000 | 0.406 | 0.000 | 0.402 | 0.666 | 0.000 | 0.469 | 0.000 |
| Customer behavior | C1 | 0.000 | 0.094 | 0.075 | 0.000 | 0.310 | 0.000 | 0.000 | 0.000 | 0.151 | 0.128 | 0.000 | 1.000 |
| | C2 | 0.000 | 0.000 | 0.009 | 0.000 | 0.039 | 0.072 | 0.000 | 0.091 | 0.000 | 0.000 | 0.000 | 0.000 |
| Policy | P1 | 0.199 | 0.402 | 0.172 | 0.000 | 0.000 | 0.187 | 0.182 | 0.155 | 0.000 | 0.000 | 0.299 | 0.000 |
| | P2 | 0.099 | 0.201 | 0.086 | 0.000 | 0.000 | 0.093 | 0.091 | 0.085 | 0.000 | 0.441 | 0.000 | 0.000 |
| | P3 | 0.000 | 0.000 | 0.000 | 0.000 | 0.000 | 0.000 | 0.000 | 0.023 | 0.000 | 0.000 | 0.000 | 0.000 |

**Table A2.** Limited super matrix.

| Barrier | Sub-Barrier | Financial | | Infrastructure | | | Technology | | Customer Behavior | | Policy | | |
|---|---|---|---|---|---|---|---|---|---|---|---|---|---|
| | | F1 | F2 | I1 | I2 | I3 | T1 | T2 | C1 | C2 | P1 | P2 | P3 |
| Financial | F1 | 0.148 | 0.148 | 0.148 | 0.148 | 0.148 | 0.148 | 0.148 | 0.148 | 0.148 | 0.148 | 0.148 | 0.148 |
| | F2 | 0.046 | 0.046 | 0.046 | 0.046 | 0.046 | 0.046 | 0.046 | 0.046 | 0.046 | 0.046 | 0.046 | 0.046 |
| Infrastructure | I1 | 0.065 | 0.065 | 0.065 | 0.065 | 0.065 | 0.065 | 0.065 | 0.065 | 0.065 | 0.065 | 0.065 | 0.065 |
| | I2 | 0.002 | 0.002 | 0.002 | 0.002 | 0.002 | 0.002 | 0.002 | 0.002 | 0.002 | 0.002 | 0.002 | 0.002 |
| | I3 | 0.005 | 0.005 | 0.005 | 0.005 | 0.005 | 0.005 | 0.005 | 0.005 | 0.005 | 0.005 | 0.005 | 0.005 |
| Technology | T1 | 0.130 | 0.130 | 0.130 | 0.130 | 0.130 | 0.130 | 0.130 | 0.130 | 0.130 | 0.130 | 0.130 | 0.130 |
| | T2 | 0.235 | 0.235 | 0.235 | 0.235 | 0.235 | 0.235 | 0.235 | 0.235 | 0.235 | 0.235 | 0.235 | 0.235 |

**Table A2.** *Cont.*

| Barrier | Sub-Barrier | Financial | | Infrastructure | | | Technology | | Customer Behavior | | Policy | | |
|---|---|---|---|---|---|---|---|---|---|---|---|---|---|
| | | F1 | F2 | I1 | I2 | I3 | T1 | T2 | C1 | C2 | P1 | P2 | P3 |
| Customer behavior | C1 | 0.036 | 0.036 | 0.036 | 0.036 | 0.036 | 0.036 | 0.036 | 0.036 | 0.036 | 0.036 | 0.036 | 0.036 |
| | C2 | 0.013 | 0.013 | 0.013 | 0.013 | 0.013 | 0.013 | 0.013 | 0.013 | 0.013 | 0.013 | 0.013 | 0.013 |
| Policy | P1 | 0.175 | 0.175 | 0.175 | 0.175 | 0.175 | 0.175 | 0.175 | 0.175 | 0.175 | 0.175 | 0.175 | 0.175 |
| | P2 | 0.143 | 0.143 | 0.143 | 0.143 | 0.143 | 0.143 | 0.143 | 0.143 | 0.143 | 0.143 | 0.143 | 0.143 |
| | P3 | 0.001 | 0.001 | 0.001 | 0.001 | 0.001 | 0.001 | 0.001 | 0.001 | 0.001 | 0.001 | 0.001 | 0.001 |

**Table A3.** Average direct relation matrix.

| Criteria | F1 | F2 | I1 | I2 | I3 | T1 | T2 | C1 | C2 | P1 | P2 | P3 |
|---|---|---|---|---|---|---|---|---|---|---|---|---|
| F1 | 0.000 | 3.556 | 2.000 | 1.444 | 1.889 | 2.111 | 2.667 | 2.333 | 1.778 | 2.111 | 2.889 | 1.444 |
| F2 | 2.000 | 0.000 | 1.889 | 1.222 | 2.111 | 2.000 | 2.333 | 2.889 | 1.778 | 2.556 | 2.889 | 1.444 |
| I1 | 2.222 | 1.889 | 0.000 | 1.000 | 1.889 | 1.778 | 2.778 | 2.333 | 2.778 | 2.111 | 2.111 | 1.667 |
| I2 | 1.667 | 1.000 | 1.000 | 0.000 | 1.111 | 1.111 | 1.222 | 1.333 | 0.778 | 1.444 | 1.667 | 1.667 |
| I3 | 1.778 | 1.556 | 1.333 | 1.333 | 0.000 | 1.889 | 1.667 | 2.111 | 1.000 | 1.444 | 1.222 | 1.222 |
| T1 | 2.444 | 2.778 | 2.111 | 1.444 | 2.000 | 0.000 | 2.667 | 2.222 | 2.333 | 1.556 | 1.667 | 1.667 |
| T2 | 2.778 | 2.778 | 2.556 | 1.889 | 2.111 | 3.111 | 0.000 | 2.444 | 2.889 | 2.111 | 2.333 | 1.778 |
| C1 | 1.556 | 2.000 | 2.333 | 1.111 | 2.000 | 2.000 | 1.778 | 0.000 | 2.222 | 2.556 | 2.444 | 1.889 |
| C2 | 2.222 | 1.778 | 2.222 | 1.222 | 1.889 | 2.667 | 2.333 | 2.778 | 0.000 | 1.556 | 1.444 | 1.222 |
| P1 | 2.444 | 3.222 | 2.667 | 2.444 | 2.000 | 1.444 | 1.444 | 2.778 | 1.889 | 0.000 | 3.222 | 3.000 |
| P2 | 2.222 | 3.222 | 2.111 | 2.444 | 1.889 | 1.889 | 1.556 | 2.667 | 1.778 | 2.889 | 0.000 | 2.333 |
| P3 | 1.667 | 1.444 | 0.778 | 1.667 | 1.000 | 1.444 | 2.000 | 1.889 | 0.889 | 2.000 | 1.889 | 0.000 |

**Table A4.** Normalized direct relation matrix.

| Criteria | F1 | F2 | I1 | I2 | I3 | T1 | T2 | C1 | C2 | P1 | P2 | P3 |
|---|---|---|---|---|---|---|---|---|---|---|---|---|
| F1 | 0.000 | 0.133 | 0.075 | 0.054 | 0.071 | 0.079 | 0.100 | 0.087 | 0.066 | 0.079 | 0.108 | 0.054 |
| F2 | 0.075 | 0.000 | 0.071 | 0.046 | 0.079 | 0.075 | 0.087 | 0.108 | 0.066 | 0.095 | 0.108 | 0.054 |
| I1 | 0.083 | 0.071 | 0.000 | 0.037 | 0.071 | 0.066 | 0.104 | 0.087 | 0.104 | 0.079 | 0.079 | 0.062 |
| I2 | 0.062 | 0.037 | 0.037 | 0.000 | 0.041 | 0.041 | 0.046 | 0.050 | 0.029 | 0.054 | 0.062 | 0.062 |
| I3 | 0.066 | 0.058 | 0.050 | 0.050 | 0.000 | 0.071 | 0.062 | 0.079 | 0.037 | 0.054 | 0.046 | 0.046 |
| T1 | 0.091 | 0.104 | 0.079 | 0.054 | 0.075 | 0.000 | 0.100 | 0.083 | 0.087 | 0.058 | 0.062 | 0.062 |
| T2 | 0.104 | 0.104 | 0.095 | 0.071 | 0.079 | 0.116 | 0.000 | 0.091 | 0.108 | 0.079 | 0.087 | 0.066 |
| C1 | 0.058 | 0.075 | 0.087 | 0.041 | 0.075 | 0.075 | 0.066 | 0.000 | 0.083 | 0.095 | 0.091 | 0.071 |
| C2 | 0.083 | 0.066 | 0.083 | 0.046 | 0.071 | 0.100 | 0.087 | 0.104 | 0.000 | 0.058 | 0.054 | 0.046 |
| P1 | 0.091 | 0.120 | 0.100 | 0.091 | 0.075 | 0.054 | 0.054 | 0.104 | 0.071 | 0.000 | 0.120 | 0.112 |
| P2 | 0.083 | 0.120 | 0.079 | 0.091 | 0.071 | 0.071 | 0.058 | 0.100 | 0.066 | 0.108 | 0.000 | 0.087 |
| P3 | 0.062 | 0.054 | 0.029 | 0.062 | 0.037 | 0.054 | 0.075 | 0.071 | 0.033 | 0.075 | 0.071 | 0.000 |

**Table A5.** Total relation matrix.

| Criteria | F1 | F2 | I1 | I2 | I3 | T1 | T2 | C1 | C2 | P1 | P2 | P3 |
|---|---|---|---|---|---|---|---|---|---|---|---|---|
| F1 | 0.378 | 0.539 | 0.423 | 0.339 | 0.400 | 0.430 | 0.460 | 0.505 | 0.402 | 0.447 | 0.494 | 0.373 |
| F2 | 0.430 | 0.401 | 0.404 | 0.319 | 0.392 | 0.410 | 0.432 | 0.503 | 0.386 | 0.444 | 0.476 | 0.360 |
| I1 | 0.430 | 0.457 | 0.331 | 0.305 | 0.378 | 0.397 | 0.440 | 0.477 | 0.412 | 0.421 | 0.442 | 0.359 |
| I2 | 0.278 | 0.281 | 0.241 | 0.169 | 0.234 | 0.247 | 0.259 | 0.294 | 0.225 | 0.269 | 0.289 | 0.247 |
| I3 | 0.322 | 0.343 | 0.290 | 0.245 | 0.230 | 0.311 | 0.313 | 0.364 | 0.270 | 0.308 | 0.316 | 0.265 |
| T1 | 0.438 | 0.487 | 0.404 | 0.319 | 0.383 | 0.336 | 0.439 | 0.474 | 0.399 | 0.405 | 0.429 | 0.359 |
| T2 | 0.504 | 0.548 | 0.470 | 0.376 | 0.435 | 0.491 | 0.402 | 0.543 | 0.466 | 0.476 | 0.507 | 0.410 |
| C1 | 0.396 | 0.446 | 0.398 | 0.300 | 0.370 | 0.390 | 0.395 | 0.382 | 0.382 | 0.423 | 0.439 | 0.357 |
| C2 | 0.410 | 0.431 | 0.389 | 0.295 | 0.361 | 0.406 | 0.408 | 0.468 | 0.301 | 0.383 | 0.399 | 0.327 |
| P1 | 0.481 | 0.549 | 0.461 | 0.388 | 0.421 | 0.425 | 0.442 | 0.542 | 0.421 | 0.396 | 0.527 | 0.443 |
| P2 | 0.455 | 0.528 | 0.426 | 0.373 | 0.400 | 0.421 | 0.425 | 0.516 | 0.400 | 0.473 | 0.399 | 0.405 |
| P3 | 0.322 | 0.344 | 0.275 | 0.261 | 0.268 | 0.299 | 0.325 | 0.360 | 0.268 | 0.330 | 0.342 | 0.226 |

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
