# Peer review of "Toward Net-Zero: The Barrier Analysis of Electric Vehicle Adoption and Transition Using ANP and DEMATEL"

_processes, doi:10.3390/pr10112334_

Round 1
Reviewer 1 Report
Unfortunately, we have a period of significant increase in the cost of electricity. The article deals with a very topical subject. The manuscript needs many corrections to improve its readability.
1. Introduction
"Section 3 describes MCDM solutions ..." (page 2, line 96). Before using the shortcut for the first time, it should be full name (Multi-criteria decision making).
3.1. Criterial Selection
"The adoption of electric vehicles is limited by various real and perceived barriers161616." (p. 5, l. 226). What does 161616 mean?
In Figure 2 is exactly the same as in Table 2. Only the table or only the figure is enough.
3.3. DEMATEL
"An influence map, for example, is shown in Figure 4." (p.9, l.369-370). The article does not contain Fig. 4.
4.1. ANP priority analysis
"After the criteria selection was complete, Super Decision software was used to execute the pairwise comparison process. The ANP is performed by using Super Decision software." (p. 11, l. 423-425). Repetition - twice the same.
Tables 4 and 5 should be included in the Appendix, not in the main text. Similarly, Tables 8 and 9.
"The final weights obtained from the limited super matrix are listed in Table 8." (p. 13, l. 436). It should be - Table 6.
It is not necessary to change the sub-barriers identification from F1, F2, I1 etc to C1, ..., C12. This is very confusing. Similarly in Subsection 4.2. DEMATEL priority analysis.
4.3. Result Analysis
"Table 13 11 displays the summary results of the weighting of the EVs transition barrier using the ANP and DE-MATEL methods." (p. 16, l. 484-486). You need to correct the references to the tables in subsections 4.3 and 4.4.
"Table 1 11 Result comparison" (p. 17, l. 509).
Author Response
Reply to Reviewers’ Comments
Toward Net Zero: The barrier analysis of Electric Vehicles adoption and transition by using ANP and DEMATEL
We would like to thank the reviewers for their valuable comments. The manuscript was revised according to them. Please find attached revised paper and below a detail discussion of how we responded to the questions and suggestions. In the response, content in the color of bold black is our reply and explanation about the to the reviewers’ comments and suggestions.

Reviewer 2 Report
Toward Net Zero: The barrier analysis of Electric Vehicles 2 adoption and transition by using ANP and DEMATEL
First of all, I would like to congratulate you for addressing such an important issue as greenhouse gas reduction. It is pleasing to see that solutions are being produced on this problem. When I evaluate your work from a scientific point of view, I have to write the following criticisms.
· Not every reader may be familiar with the concept of Sustainable Development Goals. A little more detail should be given on the SDG.
· I agree with the idea that the adoption and transition process should proceed in a systematic way. However, it should be explained where it is concluded that this process can be solved by multi-criteria decision-making methods.
· Spelling errors should be corrected. Climate change migration…….
· Attention should be paid to in-text citations.
· Is the only difference between the study and other studies in the literature, revealing the relationship between barriers?
· Due to which features or benefits of ANP and DEMATEL methods used were preferred in the study? This should be clearly stated.” The ANP and DEMATEL methodologies were selected because considering the nature of the problem in the real world, each barrier definitely has an influence on each other.” I don't think this sentence explains why. A more scientific and specific reason should be presented.
· Table 1 mentions the advantages of vehicle types. In order to ensure objectivity, the disadvantages of the relevant vehicle types should also be mentioned.
· It may not be enough to only deal with the emission values of the vehicles at the time of use. The effect of these vehicles on GHG during the production phase should also be investigated.
· Barriers to the spread of EVs need to be widened. In addition, there are various studies in the literature to overcome these obstacles. Their contribution to the literature should not be overlooked. It should definitely be presented to the reader in the text. It is also conceivable to include some of the factors mentioned in these studies as barriers to this study. The following studies are some of these studies.” Site selection for EVCS in Istanbul by GIS and multi-criteria decision-making”,” Optimal Locations Determination for an Electric Vehicle Charging Infrastructure in the City of Tunis, Tunisia”,” Optimal siting of electric vehicle charging stations using pythagorean fuzzy VIKOR approach”,” A novel two stage approach for electric taxis charging station site selection” etc.
· Isn't the high maintenance cost a barrier? The scarcity of sales offices can also be considered as a barrier. Charging time is one of the main barriers. Many more such barriers could be included in the study.
· The establishment phase of pairwise comparison matrices needs to be explained in more detail. "It was made in the Super Decision program." It is not enough for a scientific article. The CR value should definitely be included in the article.
· It has been said that the questionnaire for the DEMATEL method is shown in Appendix X. However, I cannot access Appendix X. In addition, the stages of creating a questionnaire need to be detailed.
· Authors should know that every reader who reads the article is not competent in DEMATEL and ANP processes. How was the Threshold value calculated? What do the D+R and D-R values mean? What was taken into account when sorting?
· Considering the significant barrier of EV transition as presented in table 12….. Table 12 is not included in the study. In addition, Table 1 was written instead of Table 11 after Table 10. Attention should be paid to the references of tables and figures in the text.
Author Response

(The authors gave the same response as above.)

Round 2
Reviewer 1 Report
Most of the comments were correctly taken into account.
It is not necessary to change the sub-barriers identification from F1, F2, I1 etc to C1, ..., C12. This is very confusing. Similarly in Subsection 4.2. DEMATEL priority analysis.
The paragraph "Moreover, the top three most ..." still needs to be corrected. (page 12, lines 444-449)
Author Response
1. It is not necessary to change the sub-barriers identification from F1, F2, I1 etc to C1, ..., C12. This is very confusing. Similarly in Subsection 4.2. DEMATEL priority analysis.
Response. Thanks for the suggestions.
The sub-barriers identification from F1, F2, I1 etc are reserved.
2. The paragraph "Moreover, the top three most ..." still needs to be corrected. (page 12, lines 444-449)
Response. Thanks for the suggestions. The description is removed.
Reviewer 2 Report
Toward Net Zero: The barrier analysis of Electric Vehicles adoption and transition by using ANP and DEMATEL
I don't understand why the following questions are not answered.
1. Authors should know that every reader who reads the article is not competent in DEMATEL and ANP processes. How was the Threshold value calculated? What do the D+R and D-R values mean? What was taken into account when sorting?
Response.
2. Considering the significant barrier of EV transition as presented in table 12….. Table 12 is not included in the study. In addition, Table 1 was written instead of Table 11 after Table 10. Attention should be paid to the references of tables and figures in the text.
Response.
Author Response
- Authors should know that every reader who reads the article is not competent in DEMATEL and ANP processes. How was the Threshold value calculated? What do the D+R and D-R values mean? What was taken into account when sorting?
Response.
Thanks for the suggestion. The description is added.
- Considering the significant barrier of EV transition as presented in table 12….. Table 12 is not included in the study. In addition, Table 1 was written instead of Table 11 after Table 10. Attention should be paid to the references of tables and figures in the text.
Response.
The typos are corrected and revised.